# MIS-Net: A deep learning-based multi-class segmentation model for CT images

**Huawei Li** **, Changying Wang** *

College of Computer Science and Technology, Qingdao University, Qingdao City, China

* wcing@qdu.edu.cn

## Abstract

The accuracy of traditional CT image segmentation algorithms is hindered by issues such as low contrast and high noise in the images. While numerous scholars have introduced deep learning-based CT image segmentation algorithms, they still face challenges, particularly in achieving high edge accuracy and addressing pixel classification errors. To tackle these issues, this study proposes the MIS-Net (Medical Images Segment Net) model, a deep learning-based approach. The MIS-Net model incorporates multi-scale atrous convolution into the encoding and decoding structure with symmetry, enabling the comprehensive extraction of multi-scale features from CT images. This enhancement aims to improve the accuracy of lung and liver edge segmentation. In the evaluation using the COVID-19 CT Lung and Infection Segmentation dataset, the left and right lung segmentation results demonstrate that MIS-Net achieves a Dice Similarity Coefficient (DSC) of 97.61. Similarly, in the Liver Tumor Segmentation Challenge 2017 public dataset, the DSC of MIS-Net reaches 98.78.

## 1 Introduction

The process of segmenting an image into components with the same characteristics is referred to as image segmentation. CT image segmentation plays a crucial role in extracting the target region for physicians, aiding in diagnosis [1]. In the analysis of radiological lung images, the segmentation of organs in CT images is a vital initial step [2]. The objective of CT image segmentation is to extract information about organs of interest and diseased organs. Accurate segmentation is essential, as errors can lead to inaccurate information in various aspects of clinical quantification [2].

Medical research relies on various imaging modalities such as Magnetic Resonance Imaging (MRI), Computed Tomography (CT), and X-ray imaging. Qingyang Yao et al. suggested that synchronous multimodal ultrasound significantly improves the detection rate and provides more accurate classification has great potential in clinical applications [3]. These modalities enhance our understanding of normal and diseased anatomical structures, playing a key role in diagnosis and treatment planning [4]. While CT images may not have the same spatial resolution as X-ray images [5], they offer high density resolution compared to two-dimensional medical images like X-rays [6]. CT images provide clear visualizations of organs such as the liver and lungs, offering rapid imaging with high spatial resolution [7, 8]. Consequently, among the

**Data Availability Statement:** The data underlying the results presented in the study are available from https://zenodo.org/records/3757476 and https://competitions.codalab.org/competitions/17094.

**Funding:** The author(s) received no specific funding for this work.

**Competing interests:** The authors have declared that no competing interests exist.

mentioned medical imaging techniques, physicians typically prefer CT imaging due to its ability to provide more accurate anatomical information about visualized structures [9].

Current segmentation methods for CT images can be broadly categorized into six main groups: Thresholding-based Methods, Region-based Methods, Shape-based Methods, Neighborhood Anatomy Methods, Machine Learning Methods, and Deep Learning Methods. The next section focuses on deep learning based methods.

## 1.1 Deep learning methods

With the rapid advancement of deep learning, manual segmentation is becoming obsolete in the field of medical image segmentation. The convolutional neural network, being less sensitive to CT image noise, serves as an effective tool to optimize existing methods and enhance segmentation accuracy. In the pursuit of semantic segmentation, wherein the goal is to achieve pixel classification, various studies have identified the encoder-decoder structure as one of the most popular frameworks [10]. Notable structures within this category include the Fully Convolutional Network (FCN) [11], U-Net [12], Deeplab [13], and V-Net [14]. Many contemporary medical image segmentation methods are enhancements based on the U-Net architecture.

Çiçek Ö et al. [15] introduced 3D U-Net for processing three-dimensional medical images. However, due to the considerable computational load and the limited number of layers in 3D U-Net, it struggled to properly extract underlying features. V-Net addresses this issue by incorporating a residual structure, resulting in improved performance and preventing gradient vanishing to expedite network convergence. Zhang proposed the Cascade V-Net [16]. Nnformer combines Convolutional Neural Networks (CNN) and Transformer, suggesting that volume-based multi-head self-attention can reduce image complexity [17]. Unetr [18] redefines the medical image segmentation task as a sequence-to-sequence prediction problem and effectively employs Transformer to capture multiscale information. The nnunet model prioritizes preprocessing, parameter design, and postprocessing [19]. MSV-Net [20] proposes a multi-scale net that requires a convolution operation after downsampling at different scales to match the number of feature map outputs. Vnetdeepsup [21] adds a deep supervision module based on V-Net to address the challenge of slow convergence rates.

In a large number of studies on CT image segmentation there are other unsupervised or semi-supervised segmentation networks in addition to supervised segmentation networks based on encoding-decoding structures. SF Qadir et al. proposed a method for automatic segmentation of vertebrae from transverse-axial CT images based on a Deep Belief Network, which utilizes a DBN model that allows for automatic extraction of vertebral body contextual features from CT images [22]. M Ahmad et al. proposed DBN-DNN method which is based on DBN for unsupervised pre-training and supervised fine-tuning and uses blocks as the basic unit for feature learning and post-processing refinement of liver by active contouring method [23]. Binsheng he et al. proposed to use immunofluorescence staining on CTC images first, and then use convolutional neural for training to improve the segmentation accuracy [24].

The practical application of medical image segmentation is mainly used for three-dimensional reconstruction of the target region and surgical assistance [25], at present, in addition to post-processing of the segmentation results to achieve three-dimensional reconstruction, there are also based on the surface reconstruction method [26].

While automatic segmentation methods based on deep learning surpass manual segmentation in terms of speed and accuracy, certain challenges persist: (1) Low accuracy in organ edge segmentation and (2) Misclassified pixels. To address these issues, this study proposes an enhanced MIS-Net model based on encoding and decoding structures with symmetry.

The proposed innovations include:

(1)Integration of Multiple 3D Atrous Spatial Pyramid Pooling (ASPP):

To enhance the receptive field of the encoding-decoding network, we integrate multiple 3D ASPPs. This integration aims to acquire multi-scale context information, thereby reducing the pixel loss rate. During the coding stage, 3D ASPP effectively extracts multi-scale features from the input CT image and eigenmaps.

(2)Utilization of Max-Pooling Downsampling in the Coding Module:

Instead of convolution downsampling, we employ max-pooling downsampling in the coding module. This choice prioritizes edge accuracy in segmentation results, effectively reduces the computational load during network training, and accelerates model convergence.

(3)Adoption of a Mixed Training Model:

We employ a mixed training model that combines the cross-entropy loss function with the dice loss function. This approach addresses the issue of uneven sample distribution, promoting a more stable model training process. By leveraging both loss functions, we aim to achieve a balanced and effective optimization during training.

## 2 Related work

### 2.1 Medical image segmentation

Over the past few years, mainstream medical image segmentation methods have predominantly employed encoding-decoding structures. The Fully Convolutional Network (FCN), introduced by Jonathan Long et al. in 2015, stands as a pioneering application of deep learning to semantic segmentation. Building upon FCN, U-Net employs channel concatenation to achieve feature fusion, making it particularly well-suited for segmenting medical images. Unet ++ [27] further enhances U-Net by fusing U-Net structures of different scales and implementing a deep supervised mechanism, thereby improving network accuracy while significantly reducing parameter count.

In the realm of computer vision, the Vision Transformer (VIT) [28] introduces Transformer architecture. SETR [29] combines VIT with traditional neural networks for semantic segmentation. Given that many medical images are in 3D, various scholars have proposed 3D network models. 3D U-Net shares a structure similar to U-Net but takes 3D data as input and conducts corresponding 3D operations. V-Net introduces a residual structure and replaces pooling layers with convolutional layers. AGSE-VNet [30] enhances V-Net by incorporating Attention Guide (AG) filters and Squeeze and Excite (SE) modules, leading to more accurate extraction of tumor regions in MRI images.

### 2.2 Atrous spatial pyramid pooling

There are many feature extraction methods such as using the Ensembling algorithm, which aims to ensure that knowledge is passed between the layers of the network by combining many models into a single trustworthy model [31]. The Spatial Pyramid Pooling (SPP) module, introduced by Kai-Ming He in 2015 [32], addresses the issue of distortion caused by image cropping and resolves the problem of repeated feature extraction by convolutional neural networks. In the YOLOv5 model, a variation called SPPF was proposed [33], offering computational efficiency and improved model speed compared to the original SPP module. The Atrous Spatial Pyramid Pooling (ASPP) module utilizes multiple parallel atrous convolutions with different sampling rates to extract multi-scale information, which is then fused to produce the final results.

Deeplabv3 [34] presented an enhanced version of the ASPP module, incorporating a Batch Normalization (BN) layer after the atrous convolution. Additionally, global pooling is applied

to the feature map, and the number of channels is adjusted to the desired value through convolution with a 1x1 kernel.

Runrui et al. [35] proposed a Residual ASPP, featuring an attention module after each atrous convolution. The outputs of these attention modules are matrix-summed individually with the outputs of the remaining attention modules, resulting in five output feature maps. Finally, these five feature maps are stitched together, and the number of channels is adjusted using a 1x1 convolution.

# 3 Methods

## 3.1 Model

**3.1.1 MIS-Net model structure.** In this section, we present the design concept of the model and the core architecture of MIS-Net. The current CT image segmentation algorithms based on deep learning often suffer from issues such as low accuracy in organ edge segmentation and misclassified pixel points. To address these challenges, we introduce the 3D ASPP module. This module is incorporated to expand the model's perceptual field, acquiring pooling features from CT images. The integration of the 3D ASPP module aims to enhance the accuracy of organ edge segmentation and mitigate misclassification of pixel points. Leveraging the 3D ASPP module's capability to fully extract multidimensional features from the feature map not only improves the model's robustness but also increases overall segmentation accuracy.

The architecture of the MIS-Net model is illustrated in Fig 1. It is constructed based on encoding and decoding structures with symmetry, featuring an encoding module and a decoding module combined with a 3D Atrous Spatial Convolutional Pooling Pyramid. The input to MIS-Net is a sequence of CT slices, and the output is the predicted segmentation of organs. The data format processed by the MIS-Net model is represented as [N, C, D, H, W], where N denotes the number of CT slices in a single input to the network. Specifically, C represents the number of channels in the input image, and D, H, and W represent the depth, height, and width of the input image, respectively. The left part of Fig 3 shows the encoding module of the model, which consists of five convolutional layers for downsampling and feature extraction, and the right part shows the decoding module of the model, which consists of four convolutional layers for upsampling and feature fusion, where the ASPP module implements the multi-scale feature extraction of the feature map.

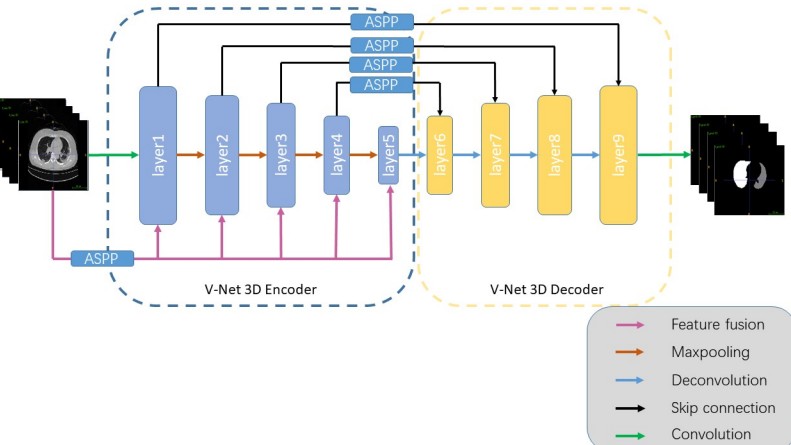

**Fig 1. MIS-Net model structure.**

*3.1.1.1 Encoding module.* The structure of the MIS-Net encoding module is depicted in Fig 2. This module comprises a series of 5x5x5 convolutional and max-pooling layers. The encoding module extracts image features through multi-layer convolution, employing the maximum pooling layer instead of the 3D convolutional layer with a kernel size of 2, as seen in the V-Net network. This choice achieves downsampling, retaining more texture information, reducing the feature map size, and expanding the number of channels.

MIS-Net integrates the 3D ASPP module into the encoding module. The 3D ASPP module fuses multi-scale features from the input CT slices, obtained through the ASPP module, into feature maps of different resolutions. This process reduces information loss and enhances edge segmentation accuracy by capturing low-level features.

Within the encoding module, a Batch Normalization (BN) layer is applied after each convolutional layer to normalize the data, mitigating the risk of gradient vanishing and overfitting to a certain extent. Additionally, the encoding module employs the Parametric Rectified Linear Unit (PReLu) activation function for de-linearization, enhancing the model's fitting ability while reducing the risk of overfitting.

*3.1.1.2 Decoding module.* The MIS-Net decoding module employs transposed convolution to restore the resolution of low-resolution feature maps. It integrates the ASPP-processed coding module features of different resolutions to capture fine-grained details lost in the compressed path, thereby enhancing the model's segmentation accuracy. The decoding module achieves an increase in the number of channels while reusing underlying features through channel stitching with the multi-scale features of the coding module obtained by ASPP. Additionally, it reduces model convergence time through ASPP residual concatenation.

To achieve feature map upsampling and recover feature map resolution, the decoding module adopts transposed convolution. This method does not require specifying an interpolation

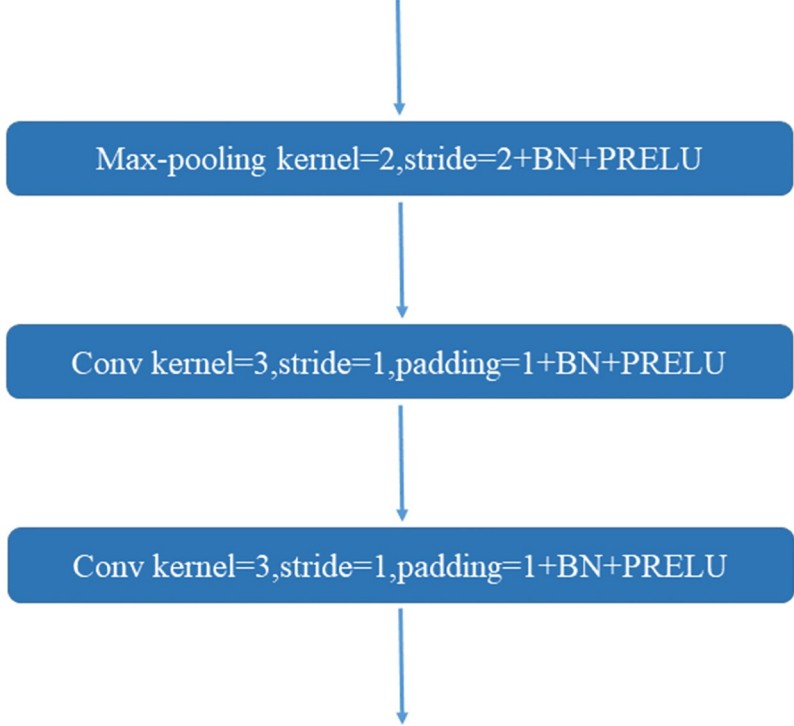

**Fig 2. MIS-Net encdoing module structure.**

method, as its parameters are learned during network training, allowing for the selection of the optimal upsampling method. Following transposed convolution, one to three convolutional layers are applied to extract more complex image features.

The final layer of the decoding module utilizes a Softmax activation function to present the CT image foreground (organ) and background classification results in a probabilistic form. The structure of the MIS-Net decoding module is illustrated in Fig 3.

**3.1.2 3D ASPP structure.** The edges of organs in CT images often exhibit blurriness, making them challenging to distinguish and leading to segmentation confusion for neural networks. In contrast, the ASPP method can capture both local and global information, integrating them into feature maps. By considering global contextual information and facilitating the fusion of multi-scale features, ASPP addresses the difficulties posed by blurred organ edges.

The 3D ASPP, comprised of 3D cavity convolutions with different expansion rates, systematically aggregates multiple scales of shape context information without compromising resolution or altering the spatial location of pixels. This allows for accurate information capture, particularly in cases involving complex edges and details. The 3D ASPP obtains multi-scale information through various perceptual field sizes, contributing to enhanced model segmentation accuracy and the resolution of organ edge segmentation. Furthermore, it achieves this with less computational intensity compared to a 3D convolutional ASPP with an equivalent perceptual field.

The MIS-Net coding module mitigates problems such as pixel misclassification and inaccurate organ edge segmentation by fusing multi-scale features obtained by 3D ASPP. While the coding module employs pooled downsampling, resulting in partial information loss, the decoding module experiences reduced feature map resolution during upsampling. MIS-Net utilizes 3D ASPP to obtain feature maps with different resolutions from the encoding module, splicing them with upsampled convolutional layer feature maps from the decoding module. This approach increases resolution, reduces pixel loss, and expands the perceptual field.

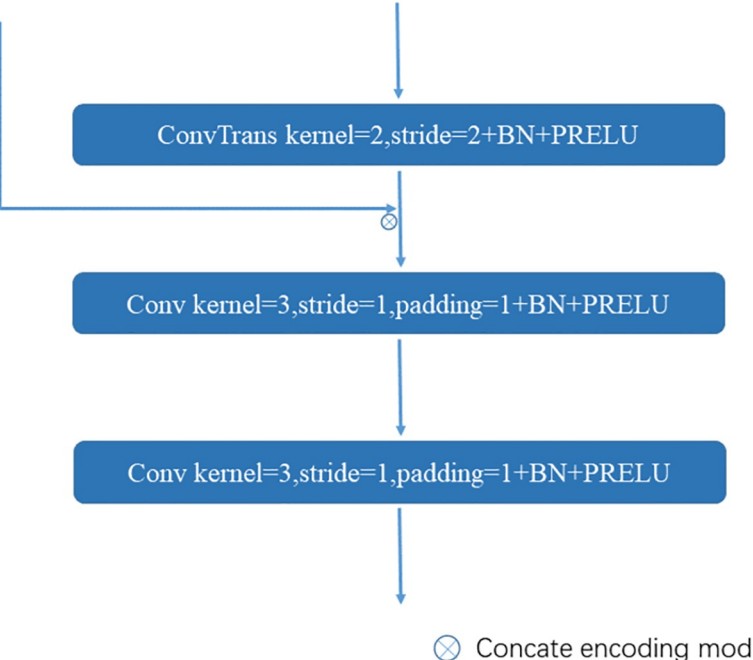

**Fig 3. MIS-Net decoding module structure.**

The 3D ASPP samples 3D cavity convolutions in parallel with different sampling rates, capturing feature context at multiple scales. The features extracted at different sampling rates are fused to generate the final result. This enhances the network's ability to acquire multi-scale features without sacrificing feature resolution while expanding the perceptual field.

The specific structure of the 3D ASPP used in this work is depicted in Fig 4, featuring a total of five parallel branches. The first, second, third, and fourth branches employ a 3D atrous convolution with a kernel size of $k = 3$ and dilation coefficients of $d = 3, 6, 12$. The last branch utilizes global pooling with a convolution kernel of 1 to adjust the number of feature map channels. The channels of the five feature maps are concatenated through the concat function, and the number of channels is then adjusted to the desired value through a 3D convolution with a kernel size of 1.

$$k_e = k + (k-1)(d-1). \tag{1}$$

$$F = k_e + 2(d-1). \tag{2}$$

$$S_{out} = \frac{S_{in} + 2p - k}{s} + 1. \tag{3}$$

The convolution kernel of 3D ASPP is calculated as shown in Eq (1), and the perceptual field of ASPP is calculated as shown in Eq (2), where $k$ is the original convolution kernal size, d is the convolution expansion rate, and $k_e$ is 3D ASPP module's atrous convolution kernals size, and $F$ is the receptive field of atrous convolution.

The output feature map size of the atrous convolution layer is calculated as Eq (3), where $S_{in}$ is input feature map size, $p$ is the padding of convolution, k is the kernal size, $s$ is the stride of convolution, and $S_{out}$ is output feature map size.

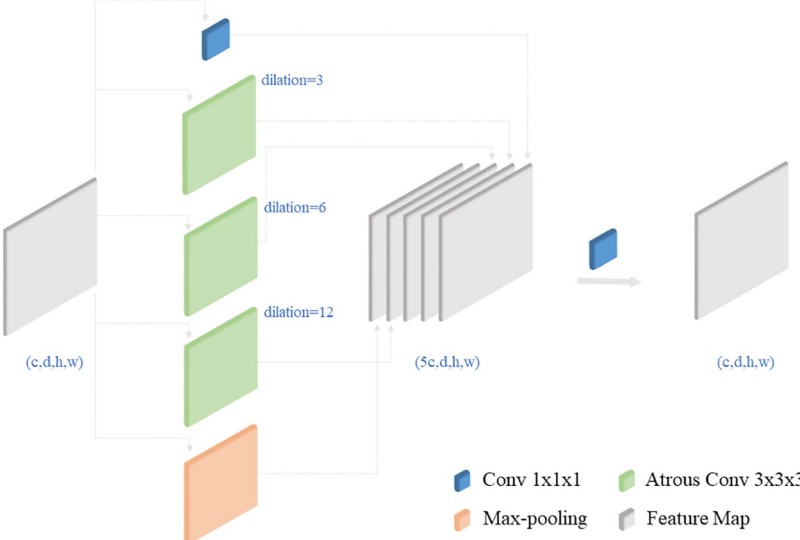

**Fig 4. 3D ASPP module structure.**

### 3.2 Loss functions

**3.2.1 DICE loss function.** The Dice loss function, proposed by Milletari et al. [14], is specifically designed to address the challenge of highly heterogeneous sample distribution in medical image segmentation. Consequently, it has become widely used in medical image processing. The Dice loss function quantifies the similarity between model predictions and true labels, defining the range of similarity within [0, 1].

Similar to the cross-entropy loss function, the Dice loss function is often utilized with the softmax activation function. This combination ensures that the sum of prediction results for multiple classifications equals 1. When a pixel belongs to the foreground of a CT image, a higher softmax value for that pixel results in a smaller Dice loss function value. Therefore, the Dice loss function helps alleviate the issue of uneven sample distribution in CT images to a certain extent. The formula for the Dice loss function is shown in Eq (4):

$$Loss = \frac{1}{batch\_size} \sum_{j=1}^{batch\_size} \sum_{i=1}^{n} \left( 1 - 2\frac{|y_{ji}^{p} \cap y_{ji}^{t}| + s}{|y_{ji}^{p}| + |y_{ji}^{t}| + s} \right). \tag{4}$$

Here, the $batch\_size$ represents the number of samples input to the network at one time, and the value of $j$ ranges from [1,batch_size]; $n$ denotes the number of categories in the samples, and the value of $i$ ranges from [1,n]. $y^p$ denotes the prediction result of the model, $y^t$ denotes the true label, and s takes the value of 1, in order to prevent the situation that the numerator denominator is 0. $|y_{ji}^{p} \cap y_{ji}^{t}|$ denotes the point multiplication operation between the predicted result and the true label first, and then the point into the result is added.

While the Dice loss function is effective, it can sometimes lead to unstable training during backpropagation. To address this, the presented work opts for a combination of the Dice loss function and the cross-entropy loss function during training. This mixed approach not only helps mitigate the problem of uneven sample distribution but also contributes to more stable training.

**3.2.2 Cross entropy loss function.** The cross-entropy loss function serves to quantify the dissimilarity between predicted and true results. It is solely dependent on the discrepancy between the predicted and true values of the model. A smaller difference between these values results in a cross-entropy loss function value closer to zero, while a larger difference yields a larger loss function value. The cross-entropy loss function is advantageous for training network models as its value is directly tied to the difference between predicted and true values. Using this loss function promotes faster convergence during the training process. Larger loss function values indicate faster learning, whereas smaller values suggest slower learning.

The calculation of the cross-entropy loss function is expressed in Eq (5). The loss function value is obtained by evaluating the error between each pixel of the prediction result and the corresponding pixel of the true label, followed by averaging all the errors.

$$Loss = \frac{1}{batch\_size} \sum_{j=1}^{batch\_size} \sum_{i=1}^{n} -y_{ji}^{p} log y_{ji}^{t} \tag{5}$$

Here, the $batch\_size$ is the number of samples input to the network at one time, and the value range of $j$ is [1,batch_size]; $n$ is the number of categories in the sample, and the value range of i is [1,n]; $y^p$ denotes the model prediction result, and $y^t$ denotes the real label.

# 4 Experiments and results

## 4.1 Data

**4.1.1 Dataset.**   The lung dataset employed in this study is sourced from the COVID-19 CT Lung and Infection Segmentation Dataset [36]. This dataset comprises CT scans from 20 patients diagnosed with COVID-19, with manual annotations of lung and infection area labels by two radiologists, validated by experienced radiologists. For model training, only the lung labels from this dataset were used, as depicted in Fig 5. Here, label 0 signifies the background, label 1 represents the left lung, and label 2 corresponds to the right lung. "Ground truth" denotes the physician-labeled label, and "Overlapping" is the image after merging the CT slice and label. The CT scans and lung labels are stored in NIFTI format. Due to the dataset's limited size, data augmentation techniques, including random cropping, random rotation, and random flipping, were applied to expand the dataset. The augmented dataset was randomly divided into two parts, with the training set containing 75% of the samples and the test set containing 25%.

For liver segmentation, the selected dataset is the Liver Tumor Segmentation Challenge 2017 (LiTS17), a public dataset specifically designed for liver segmentation. The CT slice data and annotations in LiTS17 are contributed by various clinical sites worldwide. The training dataset comprises 130 CT images with corresponding liver annotations, while the test dataset includes 70 CT images. In LiTS17, label 0 signifies the background, label 1 represents the liver, "Ground truth" indicates physician-labeled labels, and "Overlapping" represents the overlapped CT slice and labeled image. Similar to the lung dataset, the LiTS17 training set was randomly divided, with 75% of the samples allocated to training and 25% to testing, as illustrated in Fig 6.

**4.1.2 Preprocessing.**   The preprocessing steps applied to the CT images in this study are as follows:

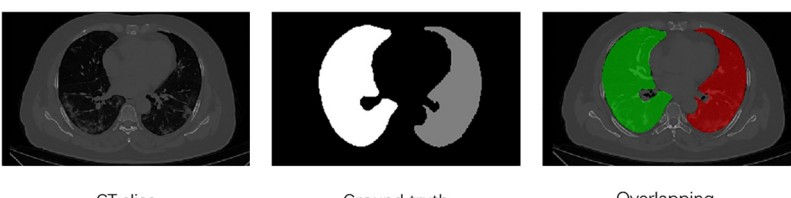

CT slice                Ground truth                Overlapping

**Fig 5. CT sections, left and right lung labels and post-overlap effects in the COVID-19 dataset.**

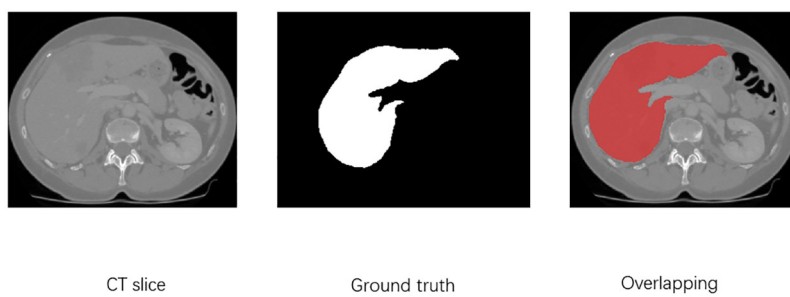

CT slice                Ground truth                Overlapping

**Fig 6. CT sections, liver labels and post-overlap effects in the LiTS dataset.**

(1)Data Normalization:

For the COVID-19 dataset, the CT values of the original CT images are cropped to the range of -1200Hu to +600Hu. This is based on the CT value range commonly used in clinical practice. The original range of the CT values in this dataset is -1000Hu to +3000Hu. For the LiTS17 dataset, the window position and window width of the original CT images are set to 30Hu and 200Hu, respectively. This effectively crops the CT values to the range of -70Hu to +130Hu. This adjustment is made to improve visibility and facilitate network learning and prediction. Subsequently, the CT values of pixel points in each dataset are normalized to the range of [0, 1]. (2)Image Cropping:

To address sample imbalance and reduce memory resource usage during model training, the CT images are resized to 128x128 pixels. The number of CT slices per CT scan is reduced to 128 to further manage memory resources. The CT slice spacing is adjusted according to the ratio of the original CT image size to the adjusted CT image size, ensuring that the real image size remains unchanged. These preprocessing steps aim to enhance the clarity of organ edges in CT images, optimize the data for effective network learning, and manage computational resources during the training process.

**4.1.3 Data enhancement.** The decision to implement data augmentation is a sound strategy, especially when dealing with a small dataset. Data augmentation is effective in expanding the effective sample size, preventing the model from overfitting to the limited dataset, and enhancing network performance. This technique is particularly valuable for training robust models with limited data.

In the experiments of this work, several data augmentation methods have been applied, including:

Random Cropping: This involves randomly selecting regions from the original CT images. It helps diversify the dataset by introducing variations in the spatial arrangement of organ tissues.

Random Flipping: This involves randomly flipping the CT images horizontally or vertically. This augmentation technique introduces mirror images, allowing the model to learn from different perspectives.

Random Rotation: This involves randomly rotating the CT images. Rotation augmentation helps the model become more invariant to different orientations of organ structures.

These data augmentation methods collectively contribute to expanding the effective dataset size while preserving the essential features of organ tissues in CT images. The augmentation process aids in preventing the model from learning irrelevant features and enhances its ability to generalize well to unseen data.

## 4.2 Evaluation metrics

The choice of evaluation metrics for model segmentation in this work reflects a comprehensive assessment of segmentation effectiveness. The selected metrics are: Dice Similarity Coefficient (DSC), Intersection over Union (IoU), Accuracy(ACC), Sensitivity and Specificity. These metrics collectively provide a thorough evaluation of the segmentation performance, considering factors such as overlap, correctness, and sensitivity. Using multiple metrics helps ensure a comprehensive understanding of the model's segmentation effectiveness.

$$DSC = \frac{2TP}{FP + 2TP + FN}. \tag{6}$$

The DSC is calculated as shown in Eq (6), where *TP* indicates the number of positive samples correctly classified, *FP* indicates the number of negative samples classified as positive, and

*FN* indicates the number of positive samples predicted as negative. The denominator of the formula can be interpreted as (*FP* + *TP*) + (*TP* + *FN*) = all samples classified as positive + all samples classified as positive.

$$IoU = \frac{TP}{FP + TP + FN}. \tag{7}$$

IoU, also known as the Jaccard index, is a standard performance measure for image segmentation. ioU, like DSC, gives the similarity between the predicted and true values, but they are calculated differently. The formula for IoU is shown in Eq (7), which indicates that IoU is the intersection of predicted and true labels divided by the concatenation between predicted and true labels.

$$ACC = \frac{TP + TN}{FP + FN + TP + TN}. \tag{8}$$

Accuracy refers to the ratio of the number of correctly predicted samples to the total number of predicted samples. The formula for ACC is shown in Eq (8).

$$Specificity = \frac{TN}{FP + TN}. \tag{9}$$

Specificity describes the probability that a category of the predicted outcome is predicted correctly, the formula for Specificity is shown in Eq (9).

$$Sensitivity = \frac{TP}{TP + FN}. \tag{10}$$

Sensitivity is also called recall and refers to the ratio of the number of correctly predicted positive samples to the total number of true positive samples, the formula for Sensitivity is shown in Eq (10).

## 4.3 Implementation details

In the experiments conducted for this work, the following hyperparameters and settings were employed: the batch size is set to 6 and the learning rate is set to 0.001. After comparing the efficiency of the Adam optimizer with that of the SGD optimizer, the Adam optimizer is used to update the weights of the neural network, and the weight decay coefficient is set to 1x10-8. The loss function used in the experiment is a hybrid loss function, in which the cross-entropy loss function and the Dice loss function are weighted as 1:1.

## 4.4 Comparison experiments

In this study, we compare the lung segmentation performance of different networks on the COVID-19 dataset, including MIS-Net, V-Net, 3d U-Net, unetr, vnetdeepsup, nnformer, Auto-model [37], MLP-Mixer [38], DALU-Net [39], Modified 3D U-Net [40] and MSV-Net. The segmentation results of the different networks are shown in Fig 7, which demonstrates the segmentation performance of Eleven networks in the validation set of four randomly selected CT scans. The segmentation results of four CT slices from the images in the validation set, where Gt represents the real labels labeled by the doctors.

Figs 7 and 8 shows that although networks such as V-Net can successfully segment the left and right lungs, there will be some mis-segmented pixel points and the edges segmented by the model are more prone to mis-segmentation. The MIS-Net proposed in this study extracts the multiscale features of the image by ASPP and fully implements feature reuse, which

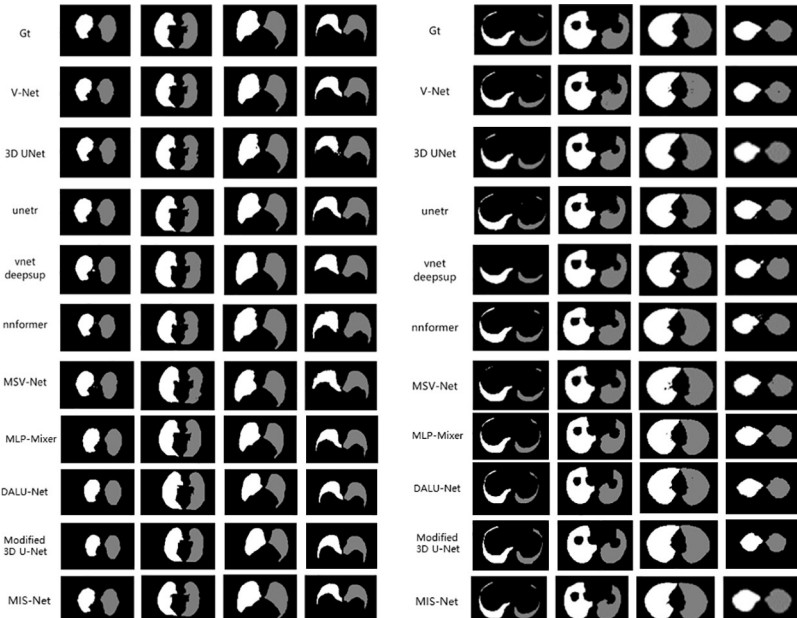

**Fig 7. Comparison of right and left lung segmentation results.**

reduces the occurrence of pixel point misclassification and improves the accuracy of lung edge segmentation. As the segmentation results of Fig 9 V-Net and MIS-Net show, the edge part of MIS-Net segmentation result is closer to the label value. Finally, in order to verify the effectiveness of this work's method, this work compares five evaluation metrics. As shown in Table 1, MIS-Net is superior in all five evaluation indexes, indicating that the MIS-Net model proposed

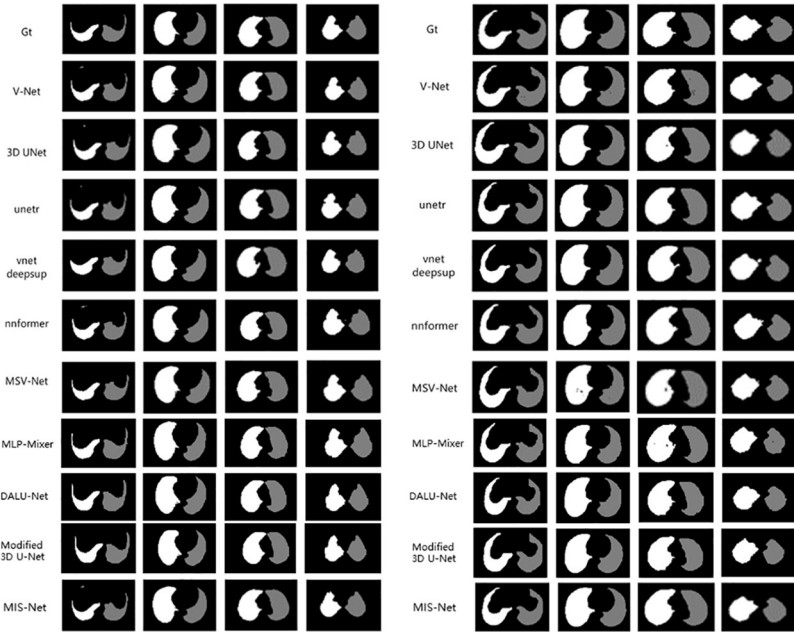

**Fig 8. Comparison of right and left lung segmentation results.**

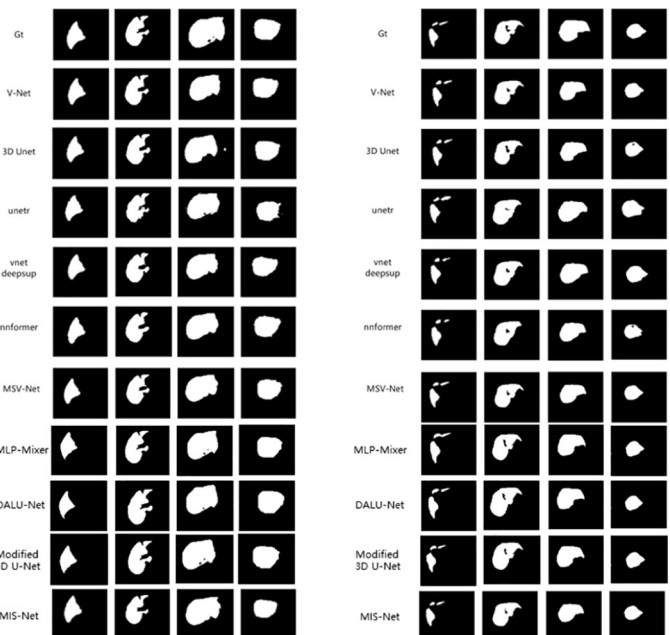

**Fig 9. Comparison of V-Net and MIS-Net CT lung segmentation details.**

in this work can segment the left and right lungs more accurately, and the segmentation performance is higher than that of the current advanced segmentation algorithms.

In this work, we compare the liver segmentation performance of Eleven networks: MIS-Net, V-Net, 3D U-Net, UNETR, VNetDeepSup, NNFormer, Auto-model, MLP-Mixer, DALU-Net, Modified 3D U-Net and MSV-Net, using the LiTS17 dataset. The segmentation results of these networks are depicted in Figs 10 and 11, illustrating their performance on four randomly selected CT images from the validation set. The segmented results correspond to four CT slices in the validation set, where Gt in Figs 10 and 11 represents the ground truth labels assigned by medical professionals.

To validate the efficacy of the MIS-Net method in liver segmentation, we compare the performance of the aforementioned networks using five evaluation metrics. As depicted in Table 2, MIS-Net outperforms the other networks in all five evaluation metrics, indicating

**Table 1. Comparison of the results of different segmentation methods for segmenting left and right lungs.**

|  | DSC | IoU | ACC | Sensitivity | Specificity |
|---|---|---|---|---|---|
| **V-Net** | 0.9638 | 0.9287 | 0.9961 | 0.9786 | 0.9987 |
| **3D Unet** | 0.9622 | 0.9204 | 0.9943 | 0.9849 | 0.9976 |
| **nnformer** | 0.9579 | 0.9160 | 0.9919 | 0.9865 | 0.9962 |
| **unetr** | 0.9658 | 0.9178 | 0.9958 | 0.9762 | 0.9988 |
| **vnet deepsup** | 0.9654 | 0.9285 | 0.9922 | 0.9582 | 0.9986 |
| **MSV-Net** | 0.9623 | 0.9231 | 0.9923 | 0.9325 | 0.9981 |
| **Auto-model** | 0.9689 | 0.9299 | 0.9962 | 0.9799 | 0.9988 |
| **MLP-Mixer** | 0.9676 | 0.9288 | 0.9958 | 0.9780 | 0.9983 |
| **DALU-Net** | 0.9693 | 0.9299 | 0.9964 | 0.9849 | 0.9985 |
| **Modified 3D U-Net** | 0.9670 | 0.9279 | 0.9953 | 0.9774 | 0.9981 |
| **MIS-Net** | **0.9761** | **0.9310** | **0.9968** | **0.9861** | **0.9993** |

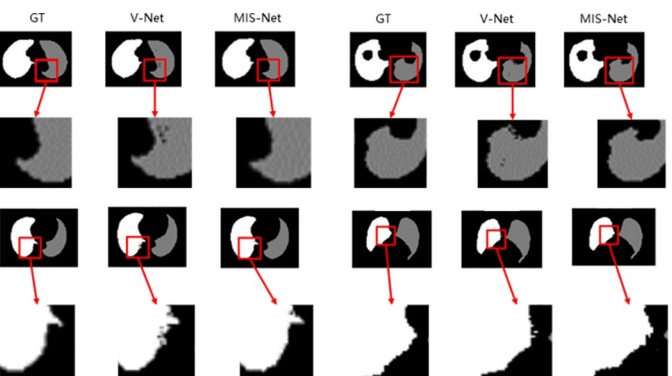

**Fig 10. Comparison of liver segmentation results.**

superior accuracy in liver segmentation. This suggests that the MIS-Net model proposed in this study provides more accurate liver segmentation, surpassing current advanced segmentation algorithms.

A more nuanced comparison of the segmentation results between V-Net and MIS-Net is illustrated in Fig 12, highlighting that the segmentation outcomes generated by MIS-Net exhibit a closer alignment with the labeled values.

## 4.5 Ablation experiments

In this study, an ablation study is conducted on the LiTS17 dataset to assess the contribution of each module in MIS-Net, as outlined in Table 3. The study encompasses the original V-Net, the V-Net with pooled downsampling, the V-Net with the added ASPP module, and the complete MIS-Net.

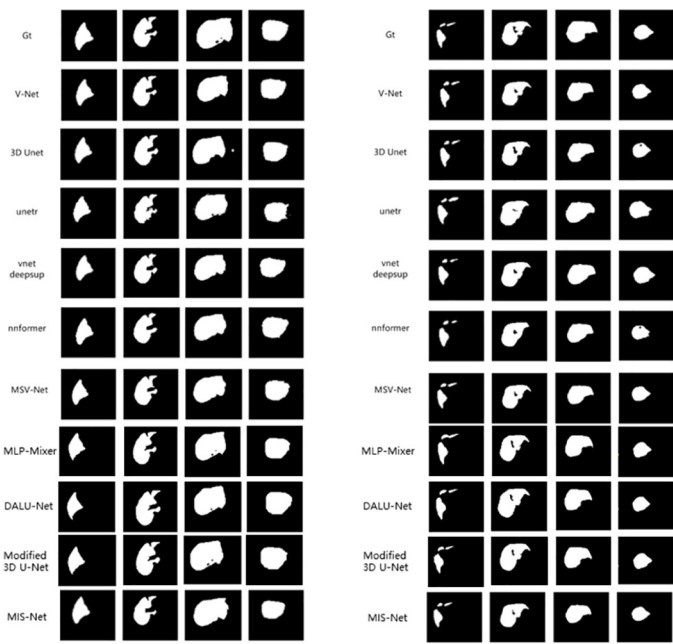

**Fig 11. Comparison of liver segmentation results.**

**Table 2. Comparison of the results of different segmentation methods for segmenting liver.**

|  | DSC | IoU | ACC | Sensitivity | Specificity |
|---|---|---|---|---|---|
| V-Net | 0.9723 | 0.9371 | 0.9968 | 0.9902 | 0.9966 |
| 3D Unet | 0.9629 | 0.9256 | 0.9963 | 0.9725 | 0.9960 |
| nnformer | 0.9619 | 0.9245 | 0.9940 | 0.9720 | 0.9951 |
| unetr | 0.9532 | 0.9184 | 0.9915 | 0.9901 | 0.9916 |
| vnet deepsup | 0.9794 | 0.9391 | 0.9944 | 0.9872 | 0.9948 |
| MSV-Net | 0.9676 | 0.9274 | 0.9957 | 0.9963 | 0.9957 |
| Auto-model | 0.9786 | 0.9389 | 0.9971 | 0.9906 | 0.9971 |
| MLP-Mixer | 0.9843 | 0.9394 | 0.9976 | 0.9909 | 0.9963 |
| DALU-Net | 0.9870 | 0.9399 | 0.9985 | 0.9912 | 0.9974 |
| Modified 3D U-Net | 0.9830 | 0.9390 | 0.9973 | 0.9907 | 0.9961 |
| **MIS-Net** | **0.9878** | **0.9415** | **0.9995** | **0.9918** | **0.9979** |

As evident from Table 3, incorporating the ASPP module slightly enhances model performance, albeit at the expense of slower convergence during training. The utilization of V-Net with a max-pooling layer instead of the convolutional downsampling layer does not substantially impact network performance. However, the pooling layer reduces the model parameters compared to the convolutional layer, effectively diminishing training time, as shown in Fig 13. Fig 13 shows the loss value comparison between V-Net model and MIS-Net training, in order to accelerate the convergence, the initial learning rate are set to 0.1, the figure only compares the first 100 epochs, as can be seen in the figure, the MIS-Net convergence speed is faster, the loss curve is smoother. Consequently, in this study, the V-Net model's convolutional downsampling is replaced by the max-pooling layer following the introduction of the 3D ASPP module. This modification aims to reduce the number of model parameters, enhance training speed, and alleviate issues related to slow convergence.

## 4.6 Three dimensional display

In this study, we employed ITK-SNAP software to achieve 3D visualization of segmentation results. The 3D visualizations for the COVID-19 dataset are presented in Fig 14, while the visualizations for LiTS17 are showcased in Fig 15. The results of the visualization demonstrate that the MIS-Net segmentation yields fewer misclassified pixel points and boasts clearer, more accurate segmentation edges.

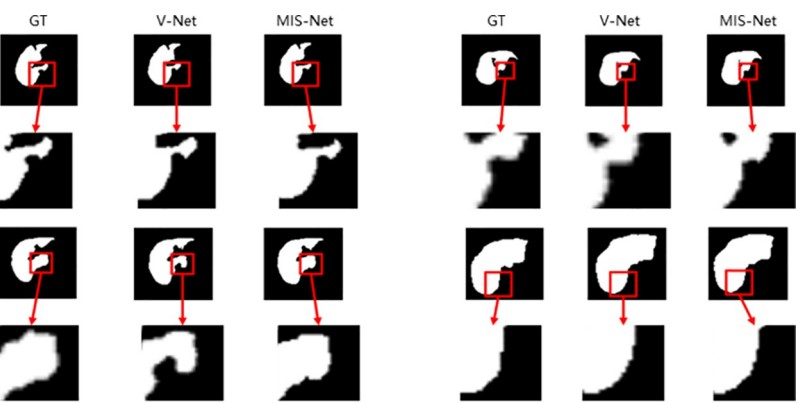

**Fig 12. Comparison of V-Net and MIS-Net CT liver segmentation details.**

**Table 3. Ablation experiments in LiTS17 datasets.**

|  | DSC | IoU | ACC | sensitivity | Specificity |
|---|---|---|---|---|---|
| **Original V-Net** | 0.9723 | 0.9371 | 0.9963 | 0.9902 | 0.9966 |
| **V-Net+ASPP** | 0.9744 | 0.9384 | 0.0.9983 | 0.9910 | 0.9969 |
| **V-Net + pool** | 0.9714 | 0.9326 | 0.9959 | 0.9900 | 0.9961 |
| **MIS-Net** | **0.9878** | **0.9415** | **0.9995** | **0.9918** | **0.9979** |

# 5 Conclusion and discussion

In this study, the MIS-Net model is introduced to address challenges related to inaccurate edge segmentation and pixel misclassification when segmenting large organs in CT images using V-Net. The MIS-Net fully extracts multi-scale features from input CT slices within the coding

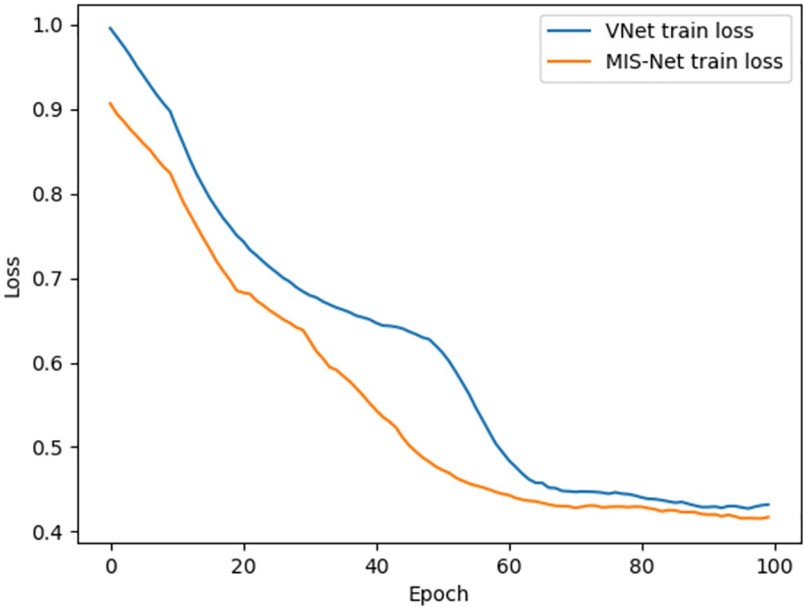

**Fig 13. VNet loss VS MIS-Net loss.**

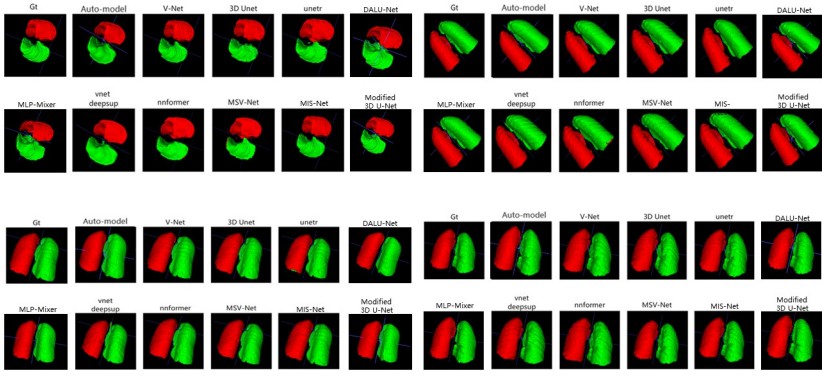

**Fig 14. COVID-19 3D visualization of segmentation results.**

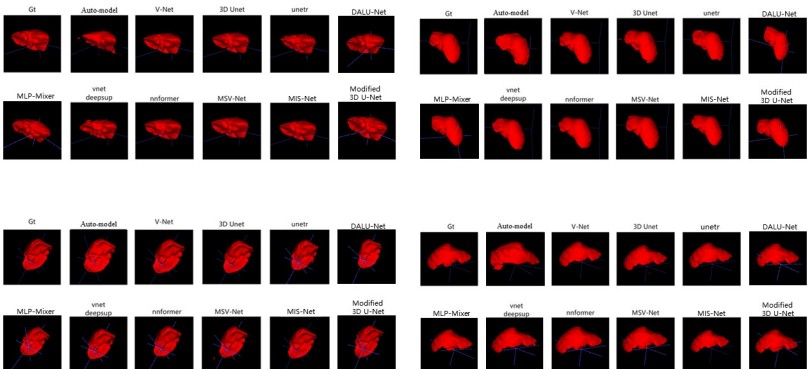

**Fig 15. LiTS17 3D visualization of segmentation results.**

module, integrating them with the feature map to mitigate feature loss and enhance the network's robustness. With the addition of the 3D ASPP module, the max-pooling layer replaces the convolutional downsampling layer in the V-Net network due to improved convergence. The combination of cross-entropy and Dice loss functions helps counter the issue of imbalanced positive and negative sample distribution in CT slices, promoting more stable model fitting by mitigating the impact of Dice loss on backpropagation.

Current clinical methods for liver and lung segmentation from abdominal CT images are cumbersome, time-consuming, and prone to bias in the boundaries of the segmentation results. And many models have poor generalization ability and cannot achieve good segmentation results in other datasets or segmentation of other target regions.

The MIS-Net proposed in this study improves the model's ability to extract multi-scale features of CT images and detect target boundaries, speeds up the convergence speed of the model, and the model can adapt to different anatomical structures and process different types of structural data, which is important for practical applications Very useful for different datasets and tasks.

Experimental comparisons between the MIS-Net model and the basic segmentation model reveal higher DSC, IoU, ACC, Sensitivity and Specificity for the MIS-Net model. These results indicate the effectiveness and rationality of the proposed approach. However, it's essential to note that the proposed method is more suitable for segmenting larger organs in CT images, such as the lung and liver. Lower segmentation accuracy is observed for smaller organs, posing challenges in precise localization and segmentation. Our future research direction is to extend the method proposed in this study so that it can be applied to segmentation of structures with smaller targets, e.g., lung nodules and liver cancer.

The challenges that the MIS-Net model proposed in this study may encounter in medical implementation mainly include: embedding the model into medical assistance systems and visually explaining its decision-making process, protecting patient data privacy, and training medical personnel to ensure their correct understanding and Use auxiliary systems, etc.

Considering the clinical application of the MIS-Net model, we will develop a medical assistance system in the future, integrate the MIS-Net model into the system, and integrate the subsequently developed models for segmenting pulmonary nodules and liver cancer into the system.

## Author Contributions

**Conceptualization:** Huawei Li.

**Data curation:** Huawei Li.

**Formal analysis:** Huawei Li, Changying Wang.

**Investigation:** Huawei Li.

**Methodology:** Huawei Li.

**Project administration:** Huawei Li, Changying Wang.

**Resources:** Huawei Li.

**Supervision:** Changying Wang.

**Validation:** Huawei Li.

**Visualization:** Huawei Li.

**Writing – original draft:** Huawei Li.

**Writing – review & editing:** Huawei Li.

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
