## [Editor Report · Decision Letter 0]

23 Nov 2023

PONE-D-23-37619MIS-Net:a deep learning-based multi-class segmentation model for CT imagesPLOS ONE

Dear Dr. Wang,

Thank you for submitting your manuscript to PLOS ONE. After careful consideration, we feel that it has merit but does not fully meet PLOS ONE’s publication criteria as it currently stands. Therefore, we invite you to submit a revised version of the manuscript that addresses the points raised during the review process.

We look forward to receiving your revised manuscript.

Kind regards,

Priyadarsan Parida, Ph.D.

Academic Editor

PLOS ONE

Journal Requirements:

Additional Editor Comments:

Reduce the Introduction Part. Only discuss about the deep learning based approaches.

What is the need of discussing the traditional approaches?

Provide a proper network architecture indicating the type of filter and the layer in details.

Compare the network with other performance parameters such as Accuracy, Sensitivity, Specificity etc.

Compare your network with other recent state of art approaches to prove the superiority of your proposed approach.

---

## [Author Response · Author response to Decision Letter 0]

13 Dec 2023

1. Reduce the Introduction Part. Only discuss about the deep learning based approaches.

We have eliminated the traditional methods from the Introduction Part, opting to focus solely on the discussion of deep learning methods（page1 and 2）.

2. Provide a proper network architecture indicating the type of filter and the layer in details.

We have added the suggested content to the manuscript on Model Part. In the encoding module and decoding module of the Model section, we've incorporated details regarding the filters and layers. Additionally, visual representations of these details are now provided in Figures 2 and 3 on pages 5 and 6, respectively

3. Compare the network with other performance parameters such as Accuracy, Sensitivity, Specificity etc.

We agree that this is a potential limitation of the study. We have supplemented the Evaluation Metrics section by introducing the metrics Accuracy, Sensitivity, and Specificity along with their respective formulas. Additionally, in the Comparison Experiments section, we have included experimental results for these three metrics in Tables 1, 2, and 3 on pages 12, 13, and 14, respectively.

4. Compare your network with other recent state of art approaches to prove the superiority of your proposed approach.

In response to the editors' suggestion, we have included a comparative experiment with the Auto-model proposed in 2022 in the Comparison Experiments section. The results of this experiment have been incorporated into Tables 1 and 2 (on pages 12 and 13) and Figures 7, 8, 11, and 12 (on pages 12 and 15).

We made our best efforts to enhance the manuscript and implemented various changes. These modifications do not affect the content or structure of the paper. In the revised version, deletions in our manuscript are highlighted in red text, and additions are marked in blue throughout the document.

---

## [Decision Letter · Decision Letter 1]

2 Jan 2024

PONE-D-23-37619R1MIS-Net:a deep learning-based multi-class segmentation model for CT imagesPLOS ONE

Dear Dr. Wang,

Thank you for submitting your manuscript to PLOS ONE. After careful consideration, we feel that it has merit but does not fully meet PLOS ONE’s publication criteria as it currently stands. Therefore, we invite you to submit a revised version of the manuscript that addresses the points raised during the review process.

We look forward to receiving your revised manuscript.

Kind regards,

Priyadarsan Parida, Ph.D.

Academic Editor

PLOS ONE

Journal Requirements:

Reviewers' comments:

Reviewer's Responses to Questions

**Comments to the Author**

1. If the authors have adequately addressed your comments raised in a previous round of review and you feel that this manuscript is now acceptable for publication, you may indicate that here to bypass the “Comments to the Author” section, enter your conflict of interest statement in the “Confidential to Editor” section, and submit your "Accept" recommendation.

Reviewer #1: All comments have been addressed

Reviewer #2: (No Response)

2. Is the manuscript technically sound, and do the data support the conclusions?

Reviewer #1: Yes

Reviewer #2: (No Response)

3. Has the statistical analysis been performed appropriately and rigorously? 

Reviewer #1: N/A

Reviewer #2: (No Response)

4. Have the authors made all data underlying the findings in their manuscript fully available?

Reviewer #1: Yes

Reviewer #2: (No Response)

5. Is the manuscript presented in an intelligible fashion and written in standard English?

Reviewer #1: Yes

Reviewer #2: (No Response)

6. Review Comments to the Author

Reviewer #1: Although the authors have done really nice job in incorporating suggested review comments but still the incorporation of the following points will help in enhancing its quality:

1. Although you have added one 2022 technique but still, I think incorporating comparison with more techniques of 2022 and 2023 (at least 3-4) will add value to your manuscript. So, please try to incorporate it. It will give a clear validation to the readers about the significance of your proposed technique.

2. There is no information given regarding the Future work for the proposed methodology. Please include it.

3. Please clearly depict the USP and significance of your research work for the chosen research problem.

4. In the References, please give more focus on new methodologies of 2022 and 2023.

Reviewer #2: In this paper, authors propose the MIS-Net (Medical Images Segment Net) model, a deep learning-based approach. The MIS-Net model incorporates multi-scale atrous convolution into the encoding and decoding structure with symmetry, enabling the comprehensive extraction of multi-scale features from CT images. This enhancement aims to improve the accuracy of lung and liver edge segmentation. Authors did a good work and interested for the readers and the following review comments are recommended, and the authors are invited to explain and modify.

1 Clinical applications of proposed MIS-Net and how?

2 Authors missed some existing prominent works of multi-class segmentation and should discuss them critically in introduction section 10.1109/ACCESS.2019.2896961; 10.1007/978-981-13-1702-6_53; 10.3390/bioengineering9110709.

3 “Effectively reduces the computational load during network training, and accelerates model convergence’, what is experimental justification of this statement?

4 Why did it need “Data Enhancement”, authors already used two datasets.

5 Authors did not mention implementation challenges.

7. PLOS authors have the option to publish the peer review history of their article (what does this mean?). If published, this will include your full peer review and any attached files.

Reviewer #1: No

Reviewer #2: No

---

## [Author Response · Author response to Decision Letter 1]

12 Jan 2024

Dear Editors and Reviewers,

On behalf of all the contributing authors, I would like to express our sincere appreciations of your letter and reviewers’ constructive comments concerning our article entitled “MIS-Net:a deep learning-based multi-class segmentation model for CT images” (Manuscript No.: PONE-D-23-37619). 

We were pleased to know that our work was rated as potentially acceptable for publication in PLOS ONE, subject to adequate revision. We thank the editors and reviewers for the time and effort that they have put into reviewing the previous version of the manuscript. Their suggestions have enabled us to improve our work. Based on the instructions provided in your letter, we uploaded the file of the revised manuscript. Accordingly, we have uploaded a copy of the original manuscript with all the changes highlighted by using the track changes mode in Latex.

These comments are all valuable and helpful for improving our article. According to editors comments, we have made extensive modifications to our manuscript and supplemented extra experiments to make our results convincing. 

Point-by-point responses to nice editors are listed below this letter.

The reviewers’ comments are laid out below in italicized font, and specific concerns have been numbered. Our response is provided in bold font.

Reviewer #1: 

1. Although you have added one 2022 technique but still, I think incorporating comparison with more techniques of 2022 and 2023 (at least 3-4) will add value to your manuscript. So, please try to incorporate it. It will give a clear validation to the readers about the significance of your proposed technique.

We have added three comparative experiments for the technologies proposed in 2022-2023 in the Comparison Experiments section of the article, which are MLP-Mixer, DALU-Net, and Modified 3D U-Net, in page 11, line 356, and page 12, lines 375-236. The comparison results are shown in Figs 7,8 and Tables 1,2 in pages 12-13.

2. There is no information given regarding the Future work for the proposed methodology. Please include it.

We are very sorry for our negligence of future work for the proposed methodology, we have added future work to the Conclusion and Discussion section, which is on page 16, line 437 of the article.

3. Please clearly depict the USP and significance of your research work for the chosen research problem.

We are very sorry for our incorrect writing USP and significance of my research work, we have corrected it in the Introduction section in page 2, lines 61-76 and added the significance of the work in the Conclusion and Discussion section in page 16, lines 426-430.

4. In the References, please give more focus on new methodologies of 2022 and 2023.

We added four 2022-2023 references to be used as comparative experiments and literature references, Refs. 27, 34, 35, 36

Reviewer #2: 

1.Clinical applications of proposed MIS-Net and how?

We have added the clinical applications of MIS-Net in Conclusion and Discussion section in page 16, lines 421-425.

2.Authors missed some existing prominent works of multi-class segmentation and should discuss them critically in introduction section 10.1109/ACCESS.2019.2896961; 10.1007/978-981-13-1702-6_53; 10.3390/bioengineering9110709.

We sincerely appreciate the valuable comments. We have checked the literature carefully and added more references on 10.1109/ACCESS.2019.2896961 and 10.1007/978-981-13-1702-6_53 into the Introduction section in the revised manuscript in page 2, lines 47-55. We have added reference on 10.3390/bioengineering9110709 into the Related Work section in page 3, lines 98-100. 

3.“Effectively reduces the computational load during network training, and accelerates model convergence’, what is experimental justification of this statement?

We added a graph comparing the training set loss curves for the first 100 epochs of VNet and MIS-Net training as a demonstration of the above problem in page 15, Fig 11.

4.Why did it need “Data Enhancement”, authors already used two datasets.

Although this article conducted model training in two data sets, the number of data samples contained in each data set is not large, of which covid-19 only includes 20 samples and lits17 only has 200 samples.

So this paper adopts the data augmentation method of rotation and flipping to solve the data shortage problem by creating more data from the available data.

5.Authors did not mention implementation challenges.

Considering the reviewer’s suggestion, we have added implementation challenges to the Conclusion and Discussion section in pages 16-17, lines 440-444.

We made our best efforts to enhance the manuscript and implemented various changes. In the revised version, deletions in our manuscript are highlighted in red text, and additions are marked in blue throughout the document.

We sincerely hope that our revised manuscript meets the standards for publication in PLOS ONE. If further revisions are necessary, please feel free to contact me at wcing@qdu.edu.cn.

We appreciate for editors’ and reviewers’ warm work earnestly, and hope that the correction will meet with approval.

Once again, thank you very much for your comments and suggestions.

Best regards,

Dr. Changying Wang

wcing@qdu.edu.cn

January 12, 2024

QingDao University

---

## [Decision Letter · Decision Letter 2]

30 Jan 2024

PONE-D-23-37619R2MIS-Net:a deep learning-based multi-class segmentation model for CT imagesPLOS ONE

Dear Dr. Wang,

Thank you for submitting your manuscript to PLOS ONE. After careful consideration, we feel that it has merit but does not fully meet PLOS ONE’s publication criteria as it currently stands. Therefore, we invite you to submit a revised version of the manuscript that addresses the points raised during the review process.

We look forward to receiving your revised manuscript.

Kind regards,

Priyadarsan Parida, Ph.D.

Academic Editor

PLOS ONE

Journal Requirements:

Reviewers' comments:

Reviewer's Responses to Questions

**Comments to the Author**

1. If the authors have adequately addressed your comments raised in a previous round of review and you feel that this manuscript is now acceptable for publication, you may indicate that here to bypass the “Comments to the Author” section, enter your conflict of interest statement in the “Confidential to Editor” section, and submit your "Accept" recommendation.

Reviewer #1: All comments have been addressed

Reviewer #2: (No Response)

2. Is the manuscript technically sound, and do the data support the conclusions?

Reviewer #1: Yes

Reviewer #2: (No Response)

3. Has the statistical analysis been performed appropriately and rigorously? 

Reviewer #1: Yes

Reviewer #2: (No Response)

4. Have the authors made all data underlying the findings in their manuscript fully available?

Reviewer #1: Yes

Reviewer #2: (No Response)

5. Is the manuscript presented in an intelligible fashion and written in standard English?

Reviewer #1: Yes

Reviewer #2: (No Response)

6. Review Comments to the Author

Reviewer #1: Dear authors, thank you for addressing all my concerns. No further suggestions from me. It can be accepted for publication.

Reviewer #2: We appreciated the authors' efforts in manuscript revision and they did a really a good work, but following concerns need to be discussed and revised carefully.

1 Authors should give detailed description of Figure 1.

2 Authors should describe what phenomenon in Figure 11.

3 An introduction is an important road map for the paper that should be consist of an opening hook to catch the researcher's attention. To make soundness of your study should include these latest related works.

I (2024). Surgical instrument posture estimation and tracking based on LSTM. ICT Express. doi: https://doi.org/10.1016/j.icte.2024.01.002

II (2023). A Novel Approach of Surface Texture Mapping for Cone-beam Computed Tomography in Image-guided Surgical Navigation. IEEE Journal of Biomedical and Health Informatics. doi: 10.1109/JBHI.2023.3298708

III (2023). Synchronous multimode ultrasound for assessing right-to-left shunt: a prospective clinical study. Frontiers in Neurology, 14. doi: 10.3389/fneur.2023.1148846

IV (2020). A New Method for CTC Images Recognition Based on Machine Learning. Frontiers in Bioengineering and Biotechnology, 8. doi: 10.3389/fbioe.2020.00897

4 Authors should be carefully check all references, volume, issue, page number etc.

7. PLOS authors have the option to publish the peer review history of their article (what does this mean?). If published, this will include your full peer review and any attached files.

Reviewer #1: **Yes: **Dr. Manju Dabass

Reviewer #2: No

---

## [Author Response · Author response to Decision Letter 2]

12 Feb 2024

Dear Editors and Reviewers,

On behalf of all the contributing authors, I would like to express our sincere appreciations of your letter and reviewers’ constructive comments concerning our article entitled “MIS-Net:a deep learning-based multi-class segmentation model for CT images” (Manuscript No.: PONE-D-23-37619). 

We were pleased to know that our work was rated as potentially acceptable for publication in PLOS ONE, subject to adequate revision. We thank the editors and reviewers for the time and effort that they have put into reviewing the previous version of the manuscript. Their suggestions have enabled us to improve our work. Based on the instructions provided in your letter, we uploaded the file of the revised manuscript. Accordingly, we have uploaded a copy of the original manuscript with all the changes highlighted by using the track changes mode in Latex.

These comments are all valuable and helpful for improving our article. According to editors comments, we have made extensive modifications to our manuscript and supplemented extra experiments to make our results convincing. 

Point-by-point responses to nice editors are listed below this letter.

The reviewers’ comments are laid out below in italicized font, and specific concerns have been numbered. Our response is provided in bold font.

Reviewer #1: 

It is a great honor to receive your approval of this work, and thank you for your comments to improve the quality of the article!

Reviewer #2: 

1. Authors should give detailed description of Figure 1.

We think this is an excellent suggestion. We have explained the change made, including the exact location where the change can be found in the revised manuscript in page 4, lines 147-151.

2. Authors should describe what phenomenon in Figure 11.

Thank you for your suggestion. As suggested by reviewer, we have added the suggested content to the manuscript on pages 14-15, lines 414-418.

3. An introduction is an important road map for the paper that should be consist of an opening hook to catch the researcher's attention. To make soundness of your study should include these latest related works

We sincerely appreciate the valuable comments. We have checked the literature carefully and added more references on 10.3389/fbioe.2020.00897, 10.1016/j.icte.2024.01.002 and 10.1109/JBHI.2023.3298708 into the Introduction section in the revised manuscript in page 2, lines 57-64. We have added reference on 10.3389/fneur.2023.1148846 into the Introduction section in page 1, lines 10-13.

4. Authors should be carefully check all references, volume, issue, page number etc.

We are very grateful to this suggestion, based on which we carefully checked all references.

We made our best efforts to enhance the manuscript and implemented various changes. In the revised version, deletions in our manuscript are highlighted in red text, and additions are marked in blue throughout the document.

We sincerely hope that our revised manuscript meets the standards for publication in PLOS ONE. If further revisions are necessary, please feel free to contact me at wcing@qdu.edu.cn.

We appreciate for editors’ and reviewers’ warm work earnestly, and hope that the correction will meet with approval.

Once again, thank you very much for your comments and suggestions.

Best regards,

Dr. Changying Wang

wcing@qdu.edu.cn

February 7, 2024

QingDao University

---

## [Decision Letter · Decision Letter 3]

21 Feb 2024

MIS-Net:a deep learning-based multi-class segmentation model for CT images

PONE-D-23-37619R3

Dear Dr. Wang,

We’re pleased to inform you that your manuscript has been judged scientifically suitable for publication and will be formally accepted for publication once it meets all outstanding technical requirements.

Kind regards,

Priyadarsan Parida, Ph.D.

Academic Editor

PLOS ONE

Additional Editor Comments (optional):

Reviewers' comments:

Reviewer's Responses to Questions

**Comments to the Author**

1. If the authors have adequately addressed your comments raised in a previous round of review and you feel that this manuscript is now acceptable for publication, you may indicate that here to bypass the “Comments to the Author” section, enter your conflict of interest statement in the “Confidential to Editor” section, and submit your "Accept" recommendation.

Reviewer #1: All comments have been addressed

Reviewer #2: (No Response)

2. Is the manuscript technically sound, and do the data support the conclusions?

Reviewer #1: Yes

Reviewer #2: (No Response)

3. Has the statistical analysis been performed appropriately and rigorously? 

Reviewer #1: N/A

Reviewer #2: (No Response)

4. Have the authors made all data underlying the findings in their manuscript fully available?

Reviewer #1: Yes

Reviewer #2: (No Response)

5. Is the manuscript presented in an intelligible fashion and written in standard English?

Reviewer #1: Yes

Reviewer #2: (No Response)

6. Review Comments to the Author

Reviewer #1: Dear authors, thank you for addressing all my concerns. There are no further suggestions from me for improving the quality of the manuscript.

Reviewer #2: We appreciate the authors' efforts in manuscript revision. They have answered all my questions satisfactorily and are highly recommended for publication.

7. PLOS authors have the option to publish the peer review history of their article (what does this mean?). If published, this will include your full peer review and any attached files.

Reviewer #1: No

Reviewer #2: No

---

## [Editor Report · Acceptance letter]

29 Feb 2024

PONE-D-23-37619R3 

PLOS ONE

Dear Dr. Wang, 

I'm pleased to inform you that your manuscript has been deemed suitable for publication in PLOS ONE. Congratulations! Your manuscript is now being handed over to our production team.

Kind regards, 

on behalf of

Dr. Priyadarsan Parida 

Academic Editor

PLOS ONE